# Don't Look Twice: Faster Video Transformers with Run-Length Tokenization

**Rohan Choudhury** [1]    **Guanglei Zhu** [1]    **Sihan Liu** [1]    **Koichiro Niinuma** [2]

**Kris M. Kitani** [1]    **László A. Jeni** [1]

[1] Carnegie Mellon University    [2] Fujitsu Research

rchoudhu@andrew.cmu.edu

## Abstract

Transformers are slow to train on videos due to extremely large numbers of input tokens, even though many video tokens are repeated over time. Existing methods to remove such uninformative tokens either have significant overhead, negating any speedup, or require tuning for different datasets and examples. We present Run-Length Tokenization (RLT), a simple approach to speed up video transformers inspired by run-length encoding for data compression. RLT efficiently finds and removes 'runs' of patches that are repeated over time prior to model inference, then replaces them with a single patch and a positional encoding to represent the resulting token's new length. Our method is *content-aware*, requiring no tuning for different datasets, and *fast*, incurring negligible overhead. RLT yields a large speedup in training, reducing the wall-clock time to fine-tune a video transformer by 30% while matching baseline model performance. RLT also works *without any training*, increasing model throughput by 35% with only 0.1% drop in accuracy. RLT speeds up training at 30 FPS by more than $100\%$, and on longer video datasets, can reduce the token count by up to 80%. Our project page is at https://rccchoudhury.github.io/projects/rlt/.

## 1    Introduction

Vision transformers [11] have enjoyed enormous success in modeling images and videos due to their scaling properties and minimal inductive bias. Unfortunately, training these models on videos, which generally have orders of magnitude more tokens than images, is significantly more expensive. One contributing factor is that video transformers tokenize videos by splitting them into uniformly sized spatiotemporal patches [2, 3], then embed them into a latent token space. As a result, the number of tokens depends only on the video's length and resolution. Researchers are thus forced to work with very short videos (<10s), as well as significantly downsample them to low frames-per-second (FPS) and low spatial resolution.

One promising solution to this problem is to reduce the number of input tokens. Compared to language input, videos are significantly less dense in information; many works observe that videos consist mostly of redundant or uninformative tokens [15, 39, 43]. However, existing methods that aim to reduce input tokens to vision transformers have had limited adoption. Learned pruning methods [34, 52] reduce model complexity measured by GFLOPS, but either incur significant overhead during training, or require padding to handle changing numbers of tokens, negating any speed-up during training. Random masking [1, 27], though fast, decreases accuracy and thus requires more training time to match performance. Moreover, although methods like random masking and Token Merging [5] do lead to wall-clock speedups, they are not content-aware: they only remove a fixed number of

38th Conference on Neural Information Processing Systems (NeurIPS 2024).

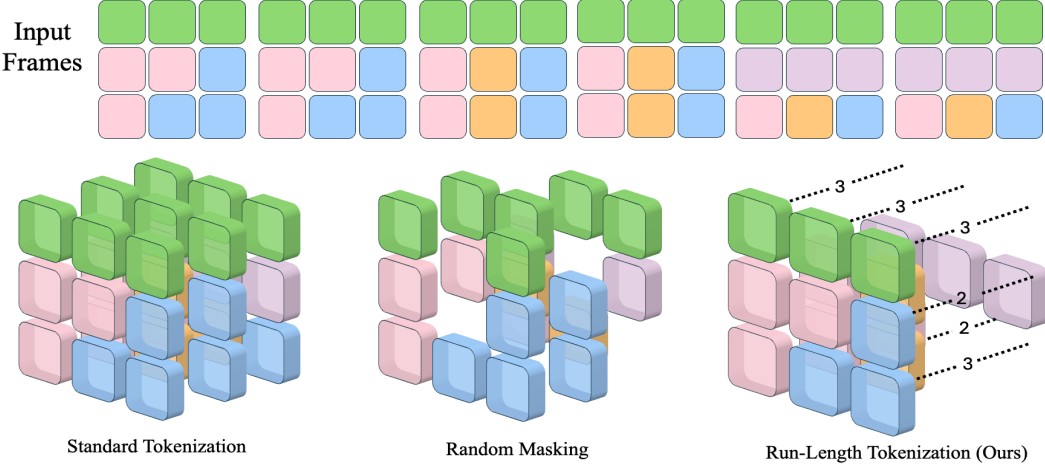

Figure 1: **Toy Example.** Given a set of input frames, with each square representing a patch, standard tokenization always produces the same number of tokens. RLT compares temporally consecutive patches and removes redundant ones, storing a single token and the run-length instead.

tokens per video, and will reduce the same number of tokens from a high-speed, high-action clip as from a still image repeated over time.

We argue that content-awareness can help more effectively reduce the number of input tokens. As an example, imagine an hour-long video of a lecture. Most of the frames are exactly the same over time, displaying a single slide. Existing methods would produce the same number of tokens from this as from an hour of motion-heavy GoPro footage, even though the two videos have significantly different amounts of content. On the other hand, video compressors, such as H.264 and H.265 [46, 41], are explicitly content-aware: rather than encoding frames independently, they encode pixel differences between consecutive frames, drastically reducing video size when there is no change.

We propose Run-Length Tokenization (RLT), which combines a simpler version of this idea with classical run-length encoding to tokenize videos for transformers. Our insight is that we can efficiently identify 'runs' of input patches that are repeated over time, enabling us to reduce the number of tokens based on the video content. When tokenizing the video, we compare consecutive patches in time and group together patches with sufficiently small differences. We then remove the "repeated" patches, and treat the remaining tokens as having variable length. Similar to how the string `aaaabb` can be run-length encoded as `a4b2`, we can add length information to each of the tokens, which incurs no additional overhead while retaining some of the information lost from removing the redundant tokens. Despite its simplicity, RLT works remarkably well - with it, we can fine-tune a video transformer in 40% faster wall-clock time than baseline ViTs while matching performance.

Our contributions are as follows: we (1) propose RLT, an alternative method to tokenize videos for vision transformers, (2) thoroughly compare its performance and compare RLT's speed to prior methods, finding significant improvements, (3) evaluate RLT's performance on high-FPS and longer videos, and (4) ablate design choices and qualitatively visualize RLT's output. We believe RLT can be a key step to significantly accelerate and further scale video understanding.

## 2 Related Work

**Video Transformers.** Vision Transformers [11] have been successfully adapted to video [2, 3, 13, 26, 38] but are generally trained and evaluated on short (<10s) video clips with relatively few frames. To efficiently handle videos, many works incorporate video-specific inductive biases in their architectures [29, 22, 56], such as memory [49, 35], compression cues [48], or modified attention mechanisms [30, 53]. Other methods, especially in video generation, project the video to a smaller latent space [20] and then split it into patches. We instead use the standard ViT formulation but apply a different tokenization scheme, reducing the number of input tokens to improve speed while maintaining performance.

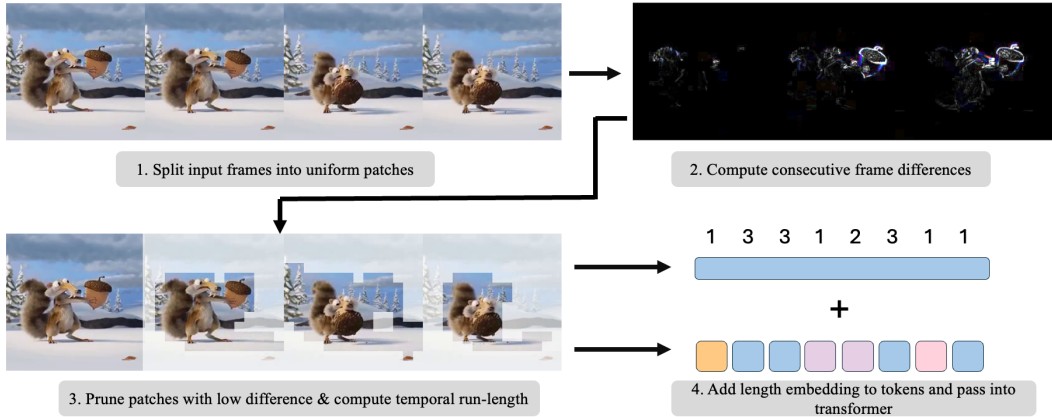

Figure 2: **RLT Overview.** RLT works by comparing temporally consecutive patches, and retaining those with L1 difference above a threshold $\tau$. The remaining tokens are augmented with a length encoding to signify their 'run-length' and passed to the transformer.

**Video Tokenization.** Prior to vision transformers, video architectures were designed to take in a fixed size input [14, 40, 55]. However, transformers can handle arbitrary numbers of input tokens [44], and training on variable-sized inputs is standard in language modeling [23]; this has been used to train vision transformers with variable resolutions [4, 10]. However, video transformers still generally use the spatiotemporal patch tokenization scheme introduced in [2, 3] which is *content-agnostic*: the number of tokens depends only on the video's length and resolution. Some works attempt to reduce input size by compressing the video to a latent space, then tokenizing [12, 33, 6], but the number of tokens still depends strictly on the input video dimensions. On the other hand, standard video compressors like HEVC [41] and AVC [46] are *content-aware*: they actively consider the differences between consecutive frames for more efficient compression. Our work applies this idea to video transformers by condensing static tokens and tracking their length.

**Faster ViTs with Fewer Tokens.** Several works have attempted to remove uninformative tokens from vision transformers. One line of work identifies such tokens either through learned modules or attention scores [34, 52, 28, 21], and prunes them at each layer. Although transformers can handle variable sized inputs, these methods require padding as token counts change unpredictably with each layer. Other works combine tokens instead of pruning them ([5, 37, 31, 51]). Most of these works require training a model for pruning or merging, with the exception of Token Merging [5], which demonstrates strong results at inference time. Inspired by the success of masked pre-training ([17, 45, 43, 15]), another line of work uses random masking to speed up training. Although masking leads to worse performance after the same number of batches, the dramatic speedup enables training for more epochs in less time [1, 10, 27, 50]. In contrast, our method matches the performance of base models with the same amount of data with large speedups, and can be stacked with random masking for even more speed benefits. Closely related to our method are EVEREST [18] and STA [36] which both exploit temporal similarity to identify redundant tokens. However, like [5] both these works require setting a constant number of tokens to remove from each video, while RLT can remove varying numbers of tokens based on the video content.

## 3   Method

Consider a vision transformer that takes as input a video $V \in \mathbb{R}^{C \times T \times H \times W}$. The standard tokenization scheme splits $V$ into a set $\mathbf{P}$ of uniformly sized, non-overlapping patches, each with size $C \times D_x \times D_y \times D_t$, with $P_t$ called the tubelet size. These patches are projected to a lower dimension $d_{embed}$ with an MLP $\mathcal{E}$, resulting in $N_P$ tokens, with each corresponding to a distinct spatiotemporal location. This results in the same number of tokens for any input video that has the same size.

In contrast, our goal is to to identify input patches that are extremely similar, then compress these redundant patches, increasing throughput and training time. Our approach is illustrated in Figure 2. In particular, we focus on *temporally consecutive* patches, those which have the same $x, y$ location

and differ by one timestep. These "static patches" correspond to visual content that does not change or move over time, and such tokens can be easily compressed.

## 3.1 Removing Static Patches

**Token Similarity.** Unlike prior works, we aim to reduce the number of total input tokens by comparing *patches* rather than tokens. By operating on patches, we do not need to run the patch embedding $\mathcal{E}$ or any layer of the model. As a result, we do not need to freeze parts of the model or propagate gradients through the pruning operation, which would require padding and negate potential speedups. This contrasts with prior works which progressively prune or combine tokens after each layer in the transformer. Furthermore, by identifying redundant patches, we can pre-compute the token distributions of various datasets and sizes of examples, allowing us to employ techniques like example-packing [23]. Finally, operating on visual patches is more interpretable and is similar to the heuristics used by video encoders [41, 46].

We next define a criterion for determining whether two consecutive patches are static. Consider two temporally consecutive patches $P_1, P_2$ that correspond to spatial location $(x, y)$ and temporal locations $t_1, t_2$ with $t_2 = t_1 + D_t$. For tubelet sizes with value $P_t > 1$, each patch consists of multiple frame crops, so that $P_1 = [P_{xy}^{t_1}, P_{xy}^{t_1+1}, ...P_{xy}^{t_1+D_t-1}]$. Given a threshold $\tau$, we consider $P_1$ and $P_2$ static if

$$\|P_{xy}^{t_2+D_t-1} - P_{xy}^{t_1}\|_1 < \tau \tag{1}$$

with $P_{xy}^{t_2+D_t-1}$ being the temporally last spatial crop of in $P_2$ and $P_{xy}^{t_1}$ the first spatial crop of $P_1$. This operation compares the "start" of the $P_1$ to the "end" of $P_2$, with the idea being that if the first crop of token $P_1$ matches the last crop of token $P_2$, the patches in between likely match as well. Notably, $\tau$ is a hyperparameter that needs to be tuned, but is *dataset-agnostic*; it simply encodes how much change between patches is allowed before they are considered different. $\tau$ controls the trade-off between speed and accuracy; while higher values reduce significantly more tokens, they treat tokens that are perceptibly different as being the same, reducing accuracy. We use $\tau > 0$ since imperceptible artifacts can occur, and follow standard procedure by running ImageNet normalization before comparing patches. We typically use $\tau = 0.1$, and provide experiments and visualizations on its effect in Section 4.3 and Appendix B.

**Pruning Procedure.** To identify all static tokens, we run the prior comparison on all pairs of temporally consecutive patches in $\mathbf{P}$ obtaining their differences and only retaining those with difference less than $\tau$. We always include the entirety of the first frame since there is no previous patch to compare it to. This results in a binary mask $M_{\text{static}}$, which we can then apply with

$$\mathbf{P}' = \mathbf{P} \circ M_{\text{static}} \tag{2}$$

with $P'$ containing $N_{P'}$ tokens and $P$ consisting of $N_P$ tokens. Note that $N_{P'} \leq N_P$ is always true; with RLT, we can never have more tokens than in the standard tokenization procedure, so the worst-case performance matches the standard vision transformer. RLT also incurs essentially no overhead as the entire process can be implemented entirely with parallelizable PyTorch [32] operations on the GPU, so training and inference are strictly faster.

The simplicity of RLT is a major advantage: in contrast to other methods, we can take advantage of transformers' ability to handle variable input sizes, and do not need to provide any additional padding. Because we make no changes to the model itself, a video transformer using RLT can make use of hardware optimizations like Flash Attention [8, 9] and memory efficient kernels [25].

Notably, the pruning procedure is *content-aware*: some videos with large amounts of static content will result in significantly fewer input tokens than videos with significant amounts of camera or subject motion. This is a desired outcome, and we discuss how to handle training with dynamic input sizes in Section 3.3.

## 3.2 Run-length Positional Encoding

Although we have reduced the number of input patches, we know that each patch represents a 'run' of static patches, with length 1 corresponding to no static content, and length $T$ corresponding to input time dimension length. Without information about the length of the 'run' of static patches, the transformer may not be able to compensate for information removed during the pruning procedure.

To address this, Bolya et al. [5] introduced Proportional Attention, which weights each token by the number of tokens in each group. On the other hand, we opt to let the model *learn* this information: we treat each token as having variable length that we can communicate through a new positional encoding. Specifically, we use a factorized encoding, described in Dehghani et al. [10], with one encoding $\phi_{xyt}$ containing positional information and the other $\phi_L$ corresponding to the length. We use a learnable length bias $\phi_L$ consisting of a single parameter of size $(T, d_{\text{embed}})$. For a given 'run' of repeated patches, we always retain the initial patch $P_{xyt}$, and thus can compute the new length $\ell_i$ as the distance from $xyt$ to the nearest 1 entry in $M_{\text{static}}$ along the $t$-axis. Concretely, for $P_{xyt}$

$$\ell_i = \min_{t'}(t' - t), \quad \text{where} \quad M_{\text{static}}(x, y, t') = 1, t' > t \tag{3}$$

This operation can also be efficiently implemented on the GPU, adding no overhead. Then, the full positional encoding becomes

$$\phi(T_i) = \phi_{xyt}(T_i) + \phi_L[\ell_i] \tag{4}$$

with the $\phi_L[\ell_i]$ representing the indexing operator. We add the positional encoding $\phi(T_i)$ to each token after running the patch embedding network $\mathcal{E}$. Unlike the pruning procedure, since we use a learnable length encoding $\phi_L$, we propagate gradients to the positional embedding, enabling the model to learn how to optimally encode variable length tokens during fine-tuning.

### 3.3 Handling Dynamic Input Sizes

Since RLT is content-aware, the number of tokens varies significantly per example. Although transformers can natively handle any input size [44], prior methods like DynamicViT [34] or A-ViT[52] produce different numbers of tokens at each layer; this requires padding or attention masking to handle batched inference during training. In our case, only the input token count is variable, but the number of tokens stays constant throughout the network, closer to the setting of NaViT [10]. Furthermore, since we know the input size before running the network, we can employ *example packing* [23], an idea from language modeling where multiple inputs with variable sizes are packed together, and tokens from individual examples attend only to each other.

At training time, the input to the transformer consists of a batch of tokenized videos, $V_1, V_2, ...V_B$, each with size $T_1, T_2, ...T_B$. Rather than pass an input $(B, \max_i T_i, d_{\text{embed}})$ to the network, we concatenate the video tensors to produce $V' = V_1 \oplus V_2 \oplus V_3...V_B$, resulting in input size $(1, \sum_{i=1}^{B} T_i, d_{\text{embed}})$. We then construct a block-diagonal attention mask so that tokens only attend to other tokens from the same video, which we add during the attention operation. Since every token in $V'$ is attending only to tokens from the same example, this does not reduce throughput and is also compatible with existing hardware-efficient attention implementations. To compute the class prediction in action recognition, we split each example out and compute its prediction as the mean of each example token, as in [43]. We then project it to dimension $N_C$, resulting in output of size $(B, N_C)$ to which we can apply standard cross-entropy losses during training.

We note that typically example packing results in a constant number of input tokens, with a variable number of input examples. A key difference between RLT and Dehghani et al. [10] is that data augmentations such as RandAugment [7] can alter the visual content and thus number of tokens of input videos, rendering greedy example packing strategies inapplicable during data loading. We opt to use a constant number of examples per GPU, with high enough batch size sufficiently reducing variance in input size.

## 4 Experimental Results

To analyze RLT's impact on performance and speed, we conduct several experiments on standard action recognition tasks. We measure the speedup on model training at several scales in Section 4.1 as well as RLT's effect as a drop-in addition at inference time in Section 4.2. We perform ablations in Section 4.3, then evaluate RLT's effect on higher FPS videos and long video datasets in Section 4.4. Finally, we provide qualitative visualizations in Section 4.5.

### 4.1 Training

In Table 1 we evaluate RLT's impact on the performance of video transformers during training and its resulting speedup. We fine-tune ViT-B and ViT-L from pre-trained VideoMAE [43, 45] checkpoints,

| | Kinetics-400 | | | Something-Something-v2 | | |
|---|---|---|---|---|---|---|
| Model | Acc | FT time(8 GPU) | Speedup | Acc | FT time(8 GPU) | Speedup |
| ViT-B | 80.1 | 14.4h | 1.0× | 70.3 | 10.1h | 1.0× |
| ToMe$_{r_{64}}$ | 80.0 | 13.4h | 1.1× | 69.7 | 9.4h | 1.1× |
| Random (0.7) | 79.2 | 10.2h | 1.4× | 69.3 | 7.2h | 1.4× |
| **RLT (Ours)** | **80.1** | **10.2h** | **1.4×** | **70.2** | **7.2h** | **1.4×** |
| ViT-L | 84.8 | 21.6h | 1.0x | 74.3 | 15.2h | 1.0× |
| ToMe | 84.4 | 18.3h | 1.2× | 74.3 | 12.9h | 1.2× |
| Random | 83.1 | 15.4h | 1.4× | 74.3 | 10.8h | 1.4× |
| **RLT (Ours)** | **84.7** | **15.4h** | **1.4×** | **74.4** | **10.8h** | **1.4×** |

Table 1: **Training results on action recognition.** RLT significantly reduces fine-tuning time with comparable performance to the baseline on both Kinetics-400 and Something-Something-v2.

comparing the speed and performance with standard tokenization, random masking, and RLT. We evaluate random masking by removing $k$ tokens, with $k$ being the mean number of tokens pruned by RLT on a given dataset. For the most fair speed comparison, all evaluated models are trained with mixed-precision, memory-efficient attention and Flash Attention where possible using an 8xH100 node, as well as the optimized data loader from AVION [54] to avoid data loading bottlenecks. We use the standard Vision Transformer rather than more complex architectures such as TimesFormer [3] or MViT [26]; we found that it was significantly simpler and more efficient, matching observations from Ryali et al. [38]. We limit our analysis to fine-tuning due to computational constraints. We compare against the baseline vision transformer, as well as Token Merging and STA [36]. We also include a random masking baseline where the masking fraction is set to the average number of tokens removed by RLT, which is a stronger baseline than using a fixed standard fraction such as 0.5.

Compared to standard tokenization, RLT achieves a speed-up of up to 40%, even with heavily optimized implementations. RLT achieves the best trade-off between performance and speed, with better performance than random masking while achieving the same speedup. This demonstrates that the choice of which tokens to remove makes a nontrivial difference, and that properly identifying redundant tokens is important.

Compared to standard tokenization, RLT achieves a speed-up of up to 40%, even with heavily optimized implementations. RLT achieves the best trade-off between performance and speed, with better performance than random masking while achieving the same speedup. In particular, RLT is much faster to train than Token Merging since it is compatible with hardware-optimized implementations such as Flash-Attention [8, 9]. Unlike random masking, RLT matches the performance of the baseline ViT after the same number of training batches, while random masking requires significantly more epochs to catch up. RLT matches baseline performance across multiple scales, indicating that RLT does not degrade performance while considerably accelerating training.

### 4.2 Inference-Time Results

Although RLT was designed to speed up training, it can be used as a drop-in replacement for standard tokenization, similar to Token Merging[5]. In Table 2 we compare the top-1 accuracy, GFLOPs and throughput with RLT to standard tokenization and Token Merging [5]. We also compare against random masking for completeness, although it is intended only for training time [27]. For the most fair comparison, we randomly mask out $P$ tokens for each example, where $P$ is the mean number of tokens used by RLT; for Kinetics-400 and SSv2 this was $P = 0.72$. We do not compare to learned pruning methods like A-ViT [52] since those only present results on images. We measure throughput in clips-per-second, with each model running on a single clip at a time. In practice, video models are evaluated on multiple temporal and spatial crops; following VideoMAE[43] we measure GFLOPs on single clip and measure accuracy with 4 temporal and 3 spatial crops.

Across model sizes, RLT consistently delivers the best tradeoff between speed and accuracy. The benefit becomes more pronounced as model size increases, as at larger parameter counts, the attention operation begins to dominate the computation. Compared to baselines, RLT is significantly faster than Token Merging and outperforms all other baselines on accuracy. Token Merging cannot make use of Flash Attention and other optimizations due to its reliance on a weighted attention operation, slowing

| | Kinetics-400 | | | | Something-Something-v2 | | | |
|---|---|---|---|---|---|---|---|---|
| Model | Acc | GFLOPS | Clips/s | Speedup | Acc | GFLOPS | Clips/s | Speedup |
| ViT-B | 80.5 | 180 | 31.4 | 1.0× | 70.8 | 180 | 31.4 | 1.0× |
| ToMe$_{r_{64}}$ | 80.4 | 131 | 34.4 | 1.09× | 69.1 | 131 | 34.4 | 1.09× |
| STA$_{r_{64}}$ | 80.4 | 131 | 34.4 | 1.09× | 69.1 | 131 | 34.4 | 1.09× |
| Random | 80.1 | 120 | **53.0** | **1.68×** | 69.3 | 120 | **53.0** | **1.68×** |
| RLT (Ours) | **80.6** | **120** | 52.6 | 1.67× | **69.8** | **120** | 52.6 | 1.67× |
| ViT-L | 84.8 | 598 | 11.5 | 1.0× | 74.3 | 598 | 11.5 | 1.0× |
| STA$_{r_{64}}$ | 80.4 | 308 | 34.4 | 1.09× | 69.1 | 308 | 34.4 | 1.09× |
| ToMe$_{r_{64}}$ | 84.3 | **285** | **19.3** | **1.68×** | 73.6 | **285** | **19.3** | **1.68×** |
| Random | 84.1 | 405 | 18.8 | 1.63× | 73.3 | 405 | 18.8 | 1.63× |
| RLT (Ours) | **84.6** | 405 | 18.71 | 1.62× | **74.1** | 405 | 18.71 | 1.62× |
| ViT-H | 86.8 | 1192 | 6.65 | 1.0× | - | - | - | - |
| ToMe$_{r_{32}}$ | 86.1 | 766 | 8.51 | 1.27× | - | - | - | - |
| STA$_{r_{64}}$ | 80.4 | **611** | 34.4 | 1.09× | - | - | - | - |
| Random | 85.1 | 816 | 9.66 | 1.45× | - | - | - | - |
| RLT (Ours) | **86.3** | 816 | **9.66** | **1.45×** | - | - | - | - |

Table 2: **Inference-only results on action recognition.** With batch size 1, RLT with $\tau = 0.1$ consistently achieves the closest performance to the baseline, comparable or faster than Token Merging or random masking. We omit ViT-H results on Something-Something-v2 due to lack of existing pre-trained checkpoints.

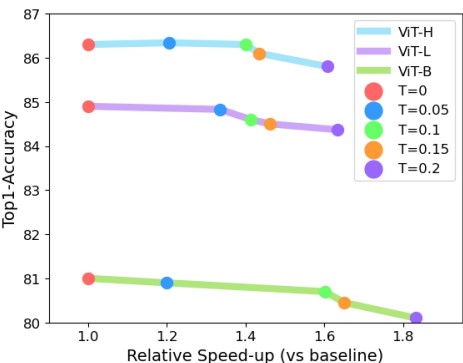

Figure 3: **Varying Difference Threshold.** When comparing the tradeoff between speedup factor and accuracy, RLT is close to baseline performance for low values of $\tau$, with a sharp drop-off after $\tau = 0.1$.

| Model | Acc | FT Time |
|---|---|---|
| ViT-B | 80.1 | 14.4h |
| RLT (no length) | 80.1 | 10.2h |
| RLT | 80.1 | 10.2h |
| RLT (no length, w/random) | 79.3 | 8.1h |
| RLT (w/random) | 79.8 | 8.1h |
| ViT-L | 84.8 | 21.6h |
| RLT (no length) | 84.6 | 15.4h |
| RLT | 84.6 | 15.4h |
| RLT (no length, w/random) | 84.2 | 11.3h |
| RLT (w/random) | 83.3 | 11.3h |

Table 3: **Effect of length encoding.** When fine-tuning with RLT only, length encoding has minimal effect, but helps significantly when combined with random masking.

it down in comparison to RLT. Although worse than RLT, random masking performs surprisingly well, likely due to the fact that most tokens in videos are redundant. Random masking can also be combined with RLT for further speed benefits, with smaller resulting performance gaps than in [27]. However, achieving the optimal performance-throughput tradeoff with random masking requires tuning for each dataset, while RLT is natively content-aware, achieving higher accuracy at similar speeds without tuning. Similarly, Token Merging [5] requires changing the $r$ parameter based on the model size and is not content aware, limiting its speed-up in highly static videos.

### 4.3 Ablations

We ablate our design choices for RLT in Figure 3 and Table 3, measuring the impact of the difference threshold and length encoding design choices at multiple model scales during training.

| Dataset | FPS | #Tokens | RLT |
|---|---|---|---|
| K400 | 7.5 | $3.8 \times 10^8$ | $2.7 \times 10^8$ (-29%) |
| K400 | 15 | $7.5 \times 10^8$ | $4.8 \times 10^8$ (-36%) |
| K400 | 30 | $1.5 \times 10^9$ | $8.2 \times 10^8$ (-45%) |
| SSv2 | 7.5 | $2.6 \times 10^8$ | $1.8 \times 10^8$ (-31%) |
| SSv2 | 15 | $5.2 \times 10^8$ | $3.2 \times 10^8$ (-38%) |
| SSv2 | 30 | $1.0 \times 10^9$ | $5.7 \times 10^8$ (-48%) |
| EK-100 | 3.5 | $1.1 \times 10^8$ | $7.2 \times 10^7$ (-36%) |
| COIN | 30 | $9.8 \times 10^9$ | $2.8 \times 10^9$ (-71%) |
| Breakfast | 15 | $1.3 \times 10^9$ | $2.7 \times 10^8$ (-79%) |

Table 4: **Per-Dataset Token Reduction.** RLT reduces tokens significantly across datasets, with higher reductions on higher FPS. On long-video datasets like COIN and Breakfast with mostly static content, RLT achieves almost 80% reduction, demonstrating its promise for scaling training.

| Model | FPS | Acc | FT Time | |
|---|---|---|---|---|
| ViT-L | 7.5 | 84.8 | 21.6h | |
| **RLT** | **7.5** | **84.6** | **15.4h** | **1.41×** |
| ViT-L | 15 | 85.8 | 45.2h | |
| **RLT** | **15** | **85.8** | **27.4h** | **1.72×** |
| ViT-L | 30 | 86.3 | 110h | |
| **RLT** | **30** | **86.2** | **52.3h** | **2.1×** |
| ViT-L | 7.5 | 74.3 | 15.1h | |
| **RLT** | **7.5** | **74.4** | **10.8h** | **1.39×** |
| ViT-L | 15 | 75.4 | 41.4h | |
| **RLT** | **15** | **75.3** | **24.1h** | **1.7×** |
| ViT-L | 30 | 76.1 | 99.8h | |
| **RLT** | **30** | **76.1** | **47.5h** | **2.0×** |

Table 5: **Training at higher FPS.** RLT enables training efficiently for higher FPS, allowing us to go beyond the standard low FPS paradigm. As FPS increases, RLT delivers larger and larger speed-ups over the baseline for training, with no decrease in accuracy.

**Difference Threshold.** The only tunable hyperparameter in RLT is the threshold $\tau$, which controls the sensitivity to change between temporally consecutive tokens. Lower values of $\tau$ indicate higher sensitivty to change. We vary $tau$ and compare the final action recognition accuracy vs. throughput and wall-clock time for several configurations, both for training and inference. These results are shown in Figure 3. We find that using $\tau = 0.1$ offered the best tradeoff in speed and performance: it matches the baseline performance while delivering a 37% speedup in training. Lower values of $\tau$ lead to similar performance, but with less of a speedup, while high values deliver larger speedups at a cost to performance. We attribute this to the existence of a 'difference cut-off': at some point, the tokens are too different to be grouped together, and the resulting tokens do not obey the assumptions made by RLT. We also note that $\tau$ is *dataset-agnostic*: it simply describes how much pixel difference is needed to consider two 16x16 patches different, and the same value of $\tau$ leads to different reductions across datasets based on the video content.

**Length Encoding.** We ablate the effect of our length encoding mechanism in Table 3. When using RLT by itself, length encoding has minimal effect. However, when combining RLT with random masking, we note a clear improvement. Due RLT's structured and predictable pruning, length encoding may be unnecessary: the transformer is able to mostly understand the length of various tokens by their associated spatial positional encoding. However, once random masking is introduced, the structure is removed, and the length encoding adds crucial information. Since including the length encoding is strictly more information and has no negative effect, we default to including it.

### 4.4 Longer Videos and Higher FPS

Standard action recognition datasets consist of short clips with downsampled FPS; an input example typically spans 2 seconds. One potential advantage of RLT is that by reducing the total number of tokens, training becomes more tractable for both longer videos and higher FPS. We evaluate the effect of training with RLT in Table 5 on action recognition datasets with higher FPS along with their training time. As before, we fine-tune these models from pre-trained VideoMAE checkpoints. Although these checkpoints were pre-trained at 7.5 FPS, we can still compare with the baseline performance to observe differences in training time or quality. Similar to the result from Table 1, we find that ViTs trained with RLT can match performance but train significantly faster, with the speed-up increasing with the FPS.

We next analyze the number of total tokens in RLT compared to the baseline for several video datasets in Table 4, including datasets with longer videos as well as higher FPS. Matching the result from Table 5, at higher FPS, RLT consistently reduces the tokens by a higher proportion. This matches

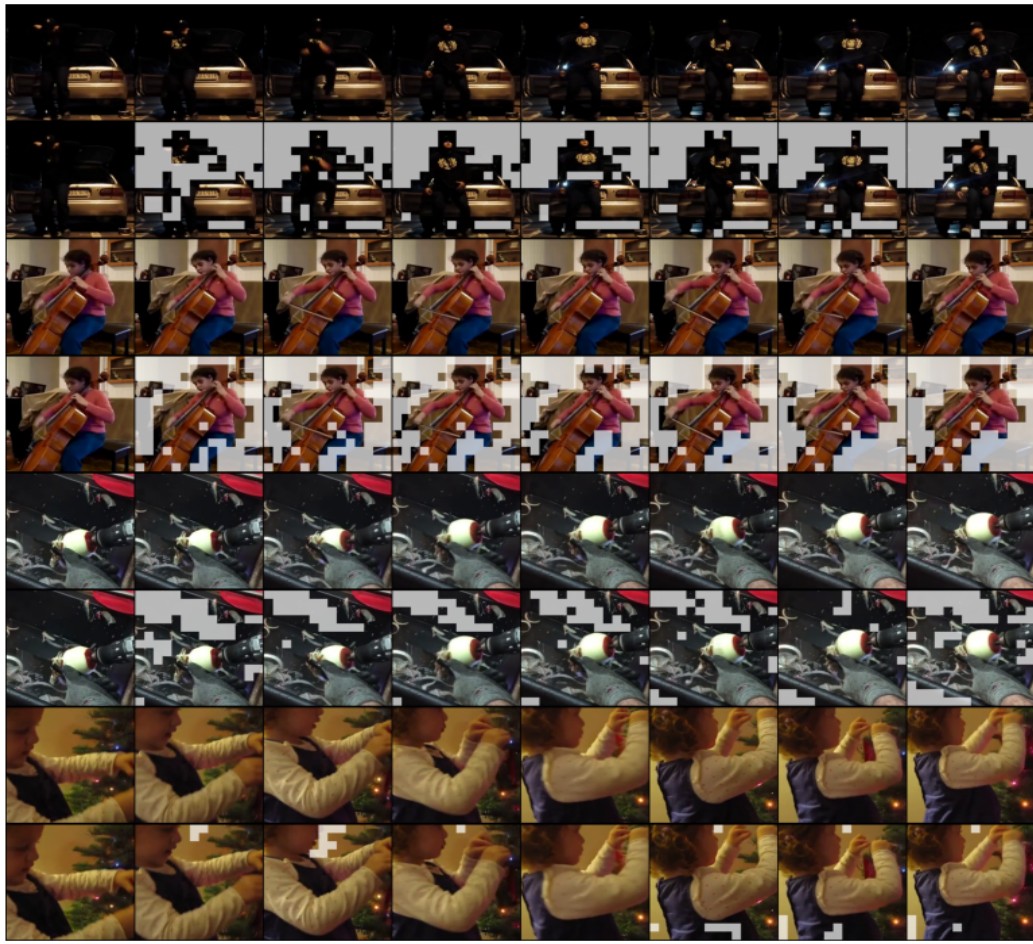

Figure 4: **Sample Visualizations.** Tokens that are compressed are visualized in gray. RLT retains tokens that change between frames while removing redundant tokens. In the top example, RLT captures the static background, and in the bottom example, due to camera motion and the motion of the girl, almost no tokens are modified. Video visualizations are available at the project page.

our intuition, since tokens between two redundant tokens at lower FPS are likely to be similar and also be removed. Furthermore, on longer video datasets, RLT can reduce the number of tokens by significantly larger margins, with reductions of up to 80% on COIN and Breakfast. These datasets in particular consist of videos filmed with fixed cameras and largely static backgrounds, demonstrating RLT's potential to drastically speed up transformers on these types of videos. Although in practice, researchers do not typically train on raw videos with large number of frames due to the heavy cost of video decoding on academic clusters, RLT presents a promising way to efficiently train on these videos at scale.

## 4.5  Visualizations

We provide some qualitative visualizations of the tokens RLT removes in Figure 4. As desired, input patches that are repeated over time are pruned by RLT. This intuitively matches with how humans often pay less attention to static tokens over time. In the top example, most of the background is black, with some motion taking place in the foreground. RLT is able to remove the constant black portions, drastically reducing the number of tokens. Similarly in the second example, RLT ensures that the tokens containing motion, with the boy's hands and instrument, are not modified, but prunes the static background. In the lower two examples, the person using the drill and the girl in the foreground move around significantly, reducing the amount of tokens that can be compressed. In such cases where there is significant subject or camera motion, RLT removes fewer tokens, resulting in similar token

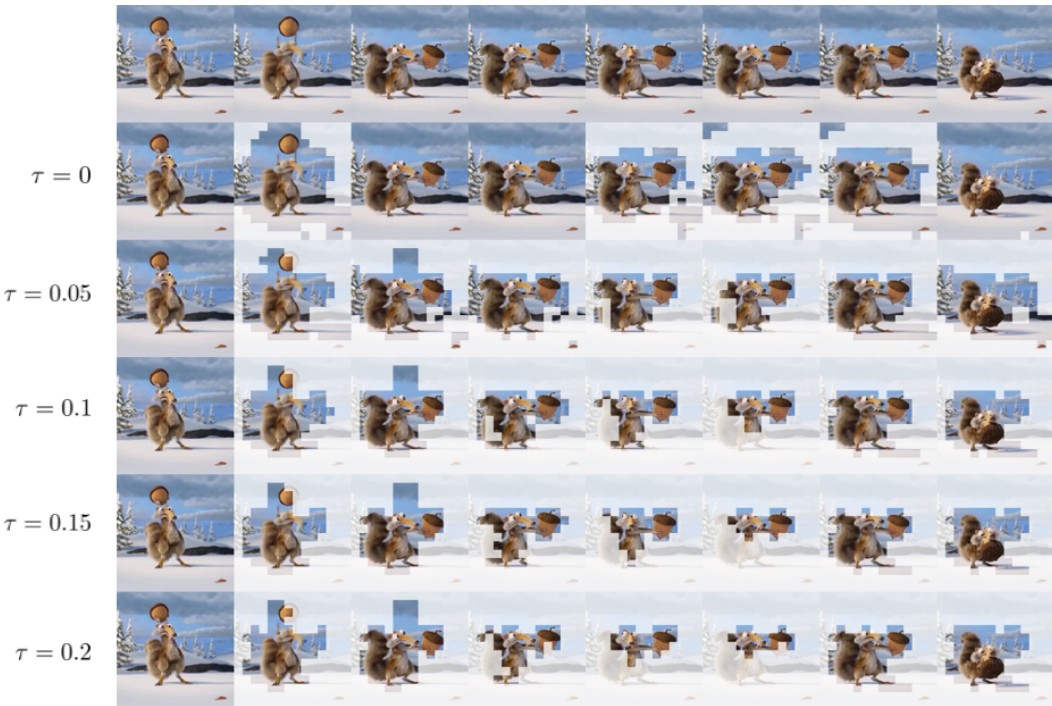

Figure 5: **Effect of** $\tau$. With low values of $\tau$, the clearest repeated patches are ablated, but imperceptible variations can prevent some visibly similar tokens from being pruned. Above $\tau = 0.1$, some tokens with slight movement are pruned.

counts to standard tokenization. However, the sensitivity of RLT to small perturbations and motion depends entirely on the $\tau$ hyperparameter. We provide further example visualizations and visualize the effect of different values of $\tau$ in Appendix B and on our project page. In Figure 5 we demonstrate the effect that the $\tau$ hyperparameter has on the input tokens. We see that as $\tau$ increases, more and more patches are included, and after $\tau = 0.1$, some patches that have change in them are pruned incorrectly. On the other hand, $\tau = 0$ includes many patches with essentially imperceptible change, which is also undesired.

## 5 Conclusion

**Summary**   We present Run-Length Tokenization (RLT), a simple alternative to standard video tokenization for video transformers that replaces temporally redundant tokens with a single token of variable length. RLT decreases transformer training and inference wall-clock time by up to 40%m achieves a better speed-accuracy tradeoff than prior works, and is simple to implement and combine with other methods. RLT demonstrates strong results during finetuning, especially at higher FPS, and even works well when applied to models without any training.

**Limitations**   Though RLT works well, it relies on a heuristic to compare temporally consecutive tokens, which could include extra tokens that are unused by the transformer. While RLT speeds up video transformers significantly, it cannot be used for dense vision tasks, such as point tracking or video generation, that require the same number of output tokens as input tokens; RLT reduces tokens before running the model and does not replace them. Furthermore, RLT does not handle camera motion well: in a video with constant camera motion, few tokens will be removed, leading to no speedup. Future work will be necessary to overcome these limitations, and we hope that RLT can inspire more research on efficient video transformers.

## Acknowledgments and Disclosure of Funding

This work is supported by Fujitsu Research of America, and RC is supported by the NSF GRFP.

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

# A   Implementation Details

Our code, demos and associated blog post are all located on our project page. In this section, we provide further details on implementation details of our experiments.

**Architecture.**   All models used were based on the `timm` [47] Vision Transformer implementation, and all fine-tuning experiments were done with pre-trained checkpoints from VideoMAE [43] and VideoMAEv2 [45]. As mentioned in 3.3, we compute output predictions for action recognition by taking the mean across the output tokens, rather than producing a separate class token.

**Baselines.**   The baselines we compared to are Token Merging [5] and random masking [27]. For all random masking experiments, we set the masking ratio $\rho$ to match the mean RLT token reduction for the given dataset. For example, on Kinetics-400 at 7.5 FPS, RLT with $\tau = 0.1$ reduces the number of tokens by $28\%$, so we randomly drop $28\%$ of the tokens during training. We use the recommended values of $r$ from the Token Merging paper, except on ViT-H, where we use $r = 32$ due to the larger depth of the model.

**Datasets.**    We train and evaluate RLT on Kinetics-400 (K400) [19] and Something-Something-v2 (SSv2) [16]. Both datasets are video classification datasets, with K400 having 400 classes and SSv2 having 174. K400 has  240k training examples and  40k test examples, while SSv2 has  170k training examples and  30k test examples. We also included experiments measuring the token reduction on the Breakfast [24] and COIN [42] datasets, both of which are smaller-scale datasets involving longer videos that range from 2-5 minutes. In particular, these datasets contain lots of fixed-camera videos with static backgrounds, leading to particularly high token reductions from RLT.

**Training Recipe.**    We do not change hyperparameters when finetuning models with different tokenization strategies, as we found the provided set to be optimal in our experiments. We follow the recommended training recipes from VideoMAE for each model size, namely training for up to 100 epochs, with batch size 256, learning rate with warm-up to $1 \times 10^{-3}$ for 5 epochs, then cosine annealing down to $1 \times 10^{-6}$. We also use RandAugment, random erasing, CutMix, and standard cropping/scaling and flipping. We do not use MixUp since it can severely affect the efficacy of RLT, and we found that removing it and only using CutMix did not affect our experiments. We also used random erasing with a single value rather than noise, enabling some of the erased tokens to be removed by RLT.

All experiments were conducted with 8xH100 Nvidia GPUs with 128 CPU cores, with 16 workers per GPU. The inference-time results were computed on a single GPU, along with the throughput and FLOPS analysis. One important detail is that data loading is often a bottleneck. We mainly relied on the fast video data loader from AVION [54], but NVIDIA DALI also works very well. However, we only recomend to use DALI on A100 or newer chips, as earlier generations have an insufficient number of dedicated decoder hardware. Each training run for the paper is specified in hours, but this does not include a few months of work testing and debugging. We used a single node for all work on this paper.

# B    More Visualizations

We include some additional visualizations here to qualitatively demonstrate which tokens RLT prunes, as well as to analyze the qualitative effect of varying the difference threshold $\tau$. In each figure, the whitened patches represent those RLT identified as static, and that are not passed to the transformer. In Figure 6, we visualize a diverse range of samples and note that RLT consistently prunes out patches that repeat across consecutive frames. One case where RLT fails to remove many tokens is the 4th example from the top, which is from a ski jumper using a GoPro; the constant camera motion means that RLT is unable to identify almost any repeated patches.

We highly encourage readers to visit our project page for video visualizations that better convey the effect of RLT.

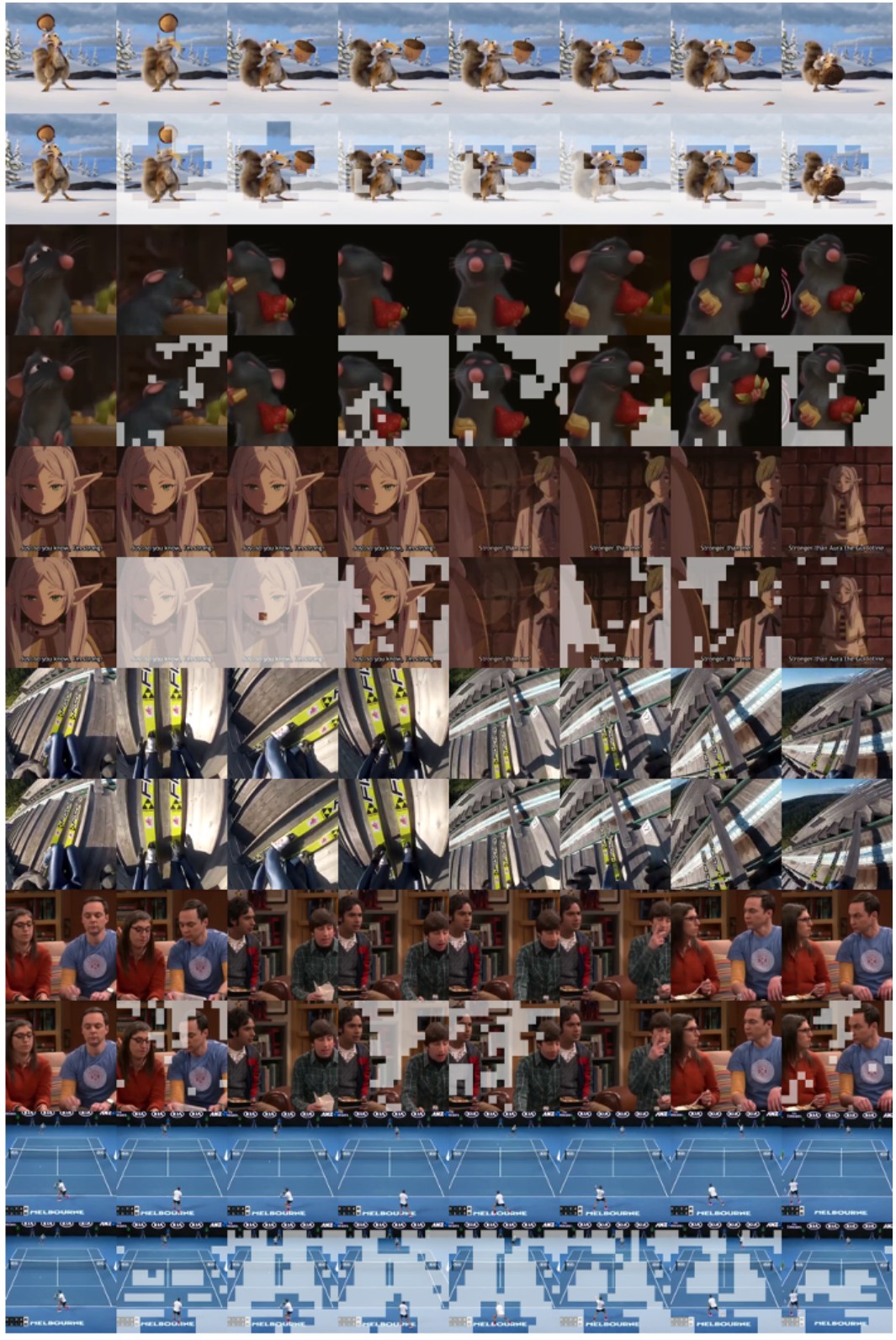

Figure 6: **More examples.** We visualize RLT's effect on videos ranging from TV shows, movies, action sequences on a GoPro, and sports. RLT consistently prunes out tokens that are repeated and static, and includes all patches that change between frames, retaining as much information as possible while cutting the number of tokens significantly.

