# OpenReview forum: "Don't Look Twice: Faster Video Transformers with Run-Length Tokenization"
_NeurIPS.cc/2024/Conference — NeurIPS 2024 spotlight_

### Official Review · Reviewer_tWGP · 2024-07-11

**Soundness:** 3
**Presentation:** 3
**Contribution:** 3
**Rating:** 4
**Confidence:** 3

**Summary:**

In this paper, the authors propose run-length tokenization, an efficient video patch merging strategy to speed up the video transformer

**Strengths:**

Please refer to Questions

**Weaknesses:**

Please refer to Questions

**Questions:**

### Strength
1. The paper is well-written and easy to follow
2. The proposed RLT is intuitive and works well

### Weakness.
1. My main concern is the experiment part is far from solid. In Tables 1 and 2, there are only two simple baselines Token Merging and random masking. Pruning is a well-studied research topic and there are many methods, the Token Merging also has many following works. Two simple baselines are not convincing. For the rest experiments, no baseline is provided. The lack of correct baselines hinders the correct evaluation of the proposed method.
2. Following 1, the token merging baseline lacks details, is it conducted within each frame? or along the temporal dimension like RLT.
3. The proposed method is somewhat too simple and intuitive, ie, merging the patches with similar pixels.
4. Some experiment results are conflicting and confusing. In Table 3, the length encoding seems nonhelpful or even harmful for the RLT, while it is claimed as a contribution of the proposed methods.

**Limitations:**

Please refer to Questions

---

> ### Author Rebuttal · Authors · 2024-08-07
>
> We deeply thank you for your helpful review and greatly appreciate your feedback. We are glad you found that our method  is “intuitive and works well”, and that the paper was “well written and easy to follow”. We address each of your concerns below.
>
> __Concern 1a: Evaluation: Two simple baselines are not enough.__
>
>  It is difficult to respond to this without direct citations of comparable pruning works or merging follow-ups. To our knowledge, the vast majority of pruning methods (e.g A-Vit [1], EVit [2], DynamicVit [3]) require training a learned token selection module. This prevents their application to out-of-the-box models while also preventing speedups during training as they require padding. Furthermore, these methods almost all only present results on images, preventing us from comparing them. On the other hand, Token Merging and random masking are two widely used methods that have been demonstrated to work well on videos, and present the fairest comparison to our work.
>
> However, the most applicable baseline we did find was Semantic-Aware Temporal Accumulation (STA) [5] which is applicable to pre-trained models out-of-the-box, but performs poorly when used during training. We re-implemented STA, optimized it further to use Flash-Attention, and then compared it to RLT in Tables 1 and 3 of the global rebuttal PDF. We find that STA is slower and performs worse in all cases than RLT. We will gladly include this result in the main text.
>
> __Concern 1b: For other experiments, no baselines are provided.__
>
> We presume that this comment refers to Tables 3, 4 and 5 of the main text. Table 3 is an ablation, and thus does not make sense to compare to other baselines. Table 4 and 5 measures the reduction in tokens in different datasets and FPS configurations. However, Token Merging, random masking and STA all reduce a constant number of tokens controlled by a single hyperparameter. Comparing the reduction from RLT to these baselines in this context does not make sense. We will reword our analysis in the paper to make this easier to follow for readers.
>
> __Concern 1c: The token merging baseline lacks details.__
>
> As stated in the Token Merging paper [4], in videos merging is conducted across both spatial and temporal dimensions. We will clarify this further in the paper.
>
> __Concern 2: The proposed method is too simple and intuitive, i.e. merging patches with similar pixels.__
>
> If a method yields significant performance improvements and is novel, it is a valuable contribution, regardless of its simplicity. In particular, Reviewers B8qa and N711 note our method is “very well motivated” and has “commendable originality”, and you note RLT’s intuitiveness in the Strengths section of the review. All three reviewers agree RLT contributes significant speedups. __Crucially, RLT does not merge patches progressively__. It finds which patches to remove before running the model. This key component enables it to remove different numbers of tokens for each input while avoiding the overhead of padding, which hobbles methods like A-Vit or DynamicVit. This point is central to the paper, and we will be sure to rewrite parts of the paper to emphasize this point better. Given our method’s novelty, simplicity and significant performance improvements, we believe that it provides an elegant solution to a problem that will be easy for many researchers to adopt in practice.
>
> __Concern 3: Some experimental results are conflicting.__
>
> We understand your concerns about the results on the learned length-encoding. As we state in the global rebuttal, supported by results in the PDF attachment, we observe no real difference with the RLT when training, but do observe a noticeable difference when combining other techniques like random masking with RLT. As we expect practitioners to combine RLT with other methods for maximizing speed, we decided to include these results to better inform the community. We will be sure to clarify the role of the learned positional embedding further in the text.
>
> Given that we have addressed the primary concerns raised in your review, we kindly request that you adjust your score.
>
> [1] Rao, Y., Zhao, W., Liu, B., Lu, J., Zhou, J. and Hsieh, C.J., 2021. Dynamicvit: Efficient vision transformers with dynamic token sparsification. Advances in neural information processing systems, 34, pp.13937-13949.
>
> [2] Liang, Y., Ge, C., Tong, Z., Song, Y., Wang, J. and Xie, P., 2022. Not all patches are what you need: Expediting vision transformers via token reorganizations. arXiv preprint arXiv:2202.07800.
>
> [3] Yin, H., Vahdat, A., Alvarez, J.M., Mallya, A., Kautz, J. and Molchanov, P., 2022. A-vit: Adaptive tokens for efficient vision transformer. In Proceedings of the IEEE/CVF conference on computer vision and pattern recognition (pp. 10809-10818).
>
> [4] Bolya, D., Fu, C.Y., Dai, X., Zhang, P., Feichtenhofer, C. and Hoffman, J., 2022. Token merging: Your vit
> but faster. International conference on learning representations.
>
> [5] Ding, S., Zhao, P., Zhang, X., Qian, R., Xiong, H. and Tian, Q., 2023. Prune spatio-temporal tokens by semantic-aware temporal accumulation. In Proceedings of the IEEE/CVF International Conference on Computer Vision (pp. 16945-16956).

---

> > ### Comment · Reviewer_tWGP · 2024-08-13
> >
> > Thanks for the efforts.
> >
> > The rebuttal solved my concern about Q1 and Q2. As for Q3, I think I hardly agree with it.
> >
> > Based on the rebuttal, I would raise my score to 4.

---

> > > ### Author Response · Authors · 2024-08-13
> > >
> > > Thanks for your response! Could you perhaps provide a litte bit more detail about your concern on Q3?
> > >
> > > To clarify our point, we found that the run-length encoding __does not__ hurt performance when combined with removing the tokens and __improves__ performance when combined with random masking. Random masking is used commonly to speed up training, and is required for techniques such as masked pre-training. Given this result, we felt it was important to include in the paper. To us, it seems overly harsh to reject the paper based on this alone.
> > >
> > > Would it be possible to raise your score to a 5? Again, we really appreciate your helpful feedback, and will gladly clarify the points in this discussion in the manuscript.

---

### Official Review · Reviewer_N711 · 2024-07-12

**Soundness:** 2
**Presentation:** 2
**Contribution:** 3
**Rating:** 7
**Confidence:** 4

**Summary:**

Current video models usually need to process every patch or tubelet of every frame, no matter if the video is very dynamic or contains patches that almost never change (e.g. static backgrounds). This submission proposes to simply omit tubelets that do not change significantly between frames, and optionally add a run-length embedding to them that signifies for how many frames that token does not change. The result is a model that only needs to process the tokens that are changing in a video, which can lead to significant speedups depending on the video content, at no loss in performance on human action recognition tasks.

**Strengths:**

- The method is very well motivated and aims to tackle an important problem, e.g. as the paper states, encoding a lecture with static background shouldn't require the same sequence length as a busy GoPro video. Dropping tubelets based on the temporal differences is better motivated than e.g. randomly dropping tokens irrespective of the video complexity.
- The RLT formulation ensures that in the worst case where every tubelet changes in every frame, the inference time and performance is as large as a non-RLT baseline (assuming the changes are greater than the threshold). With this formulation, the sequence length is quite short for static videos, and the same for very busy videos.
- The architectural modifications required are minimal, and only consist of implementing sequence packing and adding a learnable run-length embedding. This enables fine-tuning existing video models rather than having to train from scratch.
- As far as I can tell, the run-length "positional" embedding is novel
- On Kinetics-400 and Something-Something-v2, RLT can maintain performance while being significantly faster to train. Compared to a random approach that is content-unaware, RLT works better.

**Weaknesses:**

- In Table 3 it is shown that for the larger ViT-L, adding run length embeddings significantly hurts performance. On the smaller ViT-B model it achieves the same performance, but the significant performance drop on the large model calls into question how the run length encoding influences train and test dynamics, and how well tuned these ablations are.
- How well does the approach work when training from scratch compared to fine-tuning existing checkpoints? How easy is it to fine-tune in the additional run-length embedding? Would we see a similar performance drop of ViT-L as in Table 3 when training from scratch?
- Table 2 is quite messy. "ViT-B" should be greyed out, the Xs in the greyed out rows are black, and the bolding is inconsistent (ViT-L acc on Kinetics is best, and only 4 values from the entire table are bolded).
- Table 3 is not very clear. Shouldn't the second rows for each model size be called "RLT minus length" since the default RLT setting uses run length embeddings, and the third row is just "RLT"? The description of "Rand" and "length" are not very clear. The gain from combining length and rand does not seem very significant compared to the base performance.
- Table 4: How much can RLT compress the sequence length of other datasets? It should be quite cheap to evaluate this.
- It's also not clear how much influence the threshold value has on different datasets. It would be important to know if the optimal threshold value differs much between datasets, or if there is a safe range of values it can be set to.
- When training a model with RLT, should one train for the same number of tokens or same number of samples? It's not clear how much of a difference this makes between datasets.
- Is there a case where the proposed thresholding approach fails? How sensitive is it to the norm used? What if there are small but crucial differences that are below the threshold but important for the task at hand? The paper only evaluates on human action recognition tasks which could be blind to such differences.
- The method of calculating patch differences is to compute the 1-norm between the beginning and end of the two adjacent patches respectively. However there could potentially be some cases where rapid motion through a patch could conceivably be missed by this method. Did the authors test whether this was ever the case or explore alternative difference calculation methods?

**Questions:**

- The term "tokenization" is loaded with a lot of meaning, and in the introduction it's easy to assume the method performs VQ-VAE or KL-VAE like tokenization, while the method operates on spatio-temporal patches, i.e. tubeletes, that are linearly projected. I suggest making it clearer what is considered a token.
- How well would RLT perform at inference time if the run-length for all tokens is forced to be just 1, no matter the threshold?

**Limitations:**

The authors adequately address potential limitations.

---

> ### Author Rebuttal · Authors · 2024-08-07
>
> Thank you for your exceptionally thorough, helpful and clear review of our paper. We are glad you found our work to be “very well motivated”, “novel” and found it to “work better than other methods like random masking”.  We address each of your concerns individually below.
>
> __Concern 1: Learned embeddings seem to hurt performance on ViT-L.__
>
> We address this in the global rebuttal. In fact, RLT achieves about the same performance when trained with the learned embedding, and the learned embedding is especially helpful when combined with random masking.
>
>
> __Concern 2a: Did you ever test whether the thresholding would fail? There are cases where this could be a concern.__
>
> We found that a single threshold worked well for all experiments. We measured this by visually inspecting pruned patches (as in Figure 6) and measuring the performance as a function of threshold (Figure 3) However, as you note, this is an approximation for checking whether two patches are similar. We agree it could potentially affect performance for specific tasks where pixel-level details matter: for example, video generation or pose estimation. In this paper we focused on action recognition because it is the de facto task for measuring the quality of learned video representations, and our results demonstrate that RLT does not decrease the quality of these representations. As such, we believe our use of a single threshold justifies the potential tradeoff on certain downstream tasks.
>
> __Concern 2b: How sensitive is RLT to the norm used? How did this compare across datasets?__
>
> It doesn’t matter which norm is used - we found there was no appreciable difference in performance with L1 or L2. In our case, we took inspiration for using the L1 norm and a single dataset-agnostic threshold from the way video compressors [1] are implemented. Using and tuning for other types of norms could result in small differences, but L1 is simple, fast to compute, and widely used.
>
> The threshold for whether the average intensities in a block of pixels has sufficiently changed should be agnostic of the data distribution. Though the threshold value itself matters, we find that the content of the dataset videos determine the reduction from RLT itself. We include results for different values of the threshold in Table 6 of the global rebuttal PDF, supporting that the change in performance for different values of the threshold is similar for Kinetics and Something-Something v2. On the other hand, datasets with almost no motion (e.g Breakfast) will result in many pruned tokens even for extremely small thresholds.
>
> __Concern 3: How much can RLT compress the sequence length of different datasets?__
>
> We provide this analysis in Table 5 of the PDF. To be clear, each example in a dataset is compressed to a different number of tokens based on its content. We find that the average compressed sequence length does vary across datasets depending on their content; as mentioned in the main text, the Breakfast and COIN datasets have minimal motion, and RLT is able to successfully prune many more tokens.
>
> __Concern 4: Should RLT be trained for the same number of tokens or the same number of samples?__
>
> For the fairest comparison, we trained on the same number of samples. Since RLT significantly reduces the number of tokens with the same number of samples, we find that training for the same number of total tokens leads to improved performance. We demonstrate this in Table 4 of the PDF, and will include these in the final version of the paper.
>
> __Concern 5: How well does the RLT work when training from scratch? How easy is it to fine-tune the length embedding?__
>
> This is an excellent question, and one we would have liked to investigate as well. However, pre-training video transformers requires large amounts of compute that was outside of our resources. Secondly, masked pre-training (which is how all the checkpoints we fine-tuned were pretrained) already involves removing a large proportion of tokens, and thus RLT would have minimal speedup during pre-training. However, our hypothesis is that pre-training with a learned embedding would be particularly helpful during pre-training, since we noticed that training with random masking benefitted from a learned embedding, supported by Table 2 of the PDF.
>
> __Concern 6: Tables 2 and 3 are not very clear.__
>
> Thank you for pointing this out. We apologize for the lack of clarity and will gladly clean these issues up in the updated version of the manuscript. We have included fixed versions of these tables in the results PDF.
>
> __Question 1: Use of the term “tokenization" is unclear.__
>
> Thank you for bringing this to our attention. We agree that the term “tokenization” is unclear, and perhaps “patchification” would be a better alternative. We will gladly update this throughout the text to better distinguish our method from VQ-VAE and other tokenizers, as you mention.
>
> __Question 2: How would RLT perform if the length of all tokens is 1, no matter the threshold?__
>
> We are not entirely sure we understand the question, but it appears that what you refer to here is RLT without the length encoding. This result is in Table 4 of the main text, and performs about the same at inference time, and slightly worse when combined with random masking. If we have misunderstood the question, please feel free to clarify further.
>
> Given that we have addressed the primary concerns raised in your review, would you consider adjusting your score?
>
> [1] Wiegand, T., Sullivan, G.J., Bjontegaard, G. and Luthra, A., 2003. Overview of the H. 264/AVC video coding standard. IEEE Transactions on circuits and systems for video technology, 13(7), pp.560-576.

---

> > ### Author Response · Authors · 2024-08-13
> >
> > Thanks again for your detailed and constructive review. Since the discussion period is coming to a close, would it be possible to take a look at our rebuttal and let us know if you either have more questions or would adjust your score? We really appreciate your feedback, and hope we have answered your questions sufficiently.

---

> > > ### Comment · Reviewer_N711 · 2024-08-13
> > > **Official Comment by Reviewer N711**
> > >
> > > I thank the authors for their response. Most of my concerns and questions were addressed in the rebuttal, but it is still not very clear to me where the proposed thresholding approach fails. Still, I believe that the proposed method will be highly useful to the video understanding community, and I am raising my score accordingly.
> > >
> > > Regarding my question on how RLT would perform if the run length of all tokens is set to 1: The model would be trained with RLT, but at inference time one would ignore the threshold and every tubelet is only used once, and the run length is set to 1 for all of them accordingly.

---

### Official Review · Reviewer_B8qa · 2024-07-13

**Soundness:** 4
**Presentation:** 4
**Contribution:** 4
**Rating:** 8
**Confidence:** 4

**Summary:**

The authors present a compression or optimization technique applicable to video transformers for both training and inference paradigms. Through empirical evaluation, the work showcases efficiency gains achieved for fine-tuning video transformer models and also showcases inference time efficiency without any training at all, with minimal quality degradation. The code has been made available for reproducibility.

**Strengths:**

The paper introduces run length tokenization (RLT) as a mechanism to reduce static tokens in training time dynamically. RLT takes into account spatial changes with respect to the temporal aspect and effectively condense static tokens adding run length information in the input embedding. RLT draws inspiration from widely used video compression techniques such as HEVC and AVC which are content aware, making RLT also content aware as part of the input embedding of the video transformer.

The originality of the work is commendable and the presentation quality of the work is exceptional. The efficiency gains achieved using this tokenization mechanism are state of the art, in both training and inference time with no performance degradation makes this work exceptionally useful to the extended community to train and deploy video transformer models more efficiently than previous literature in the field.

**Weaknesses:**

No additional weaknesses apart from the limitations pointed out by the authors in Section 5.

**Questions:**

Out of curiosity, how did the formulation of RLT come up?

---

> ### Author Rebuttal · Authors · 2024-08-07
>
> We deeply appreciate your kind comments towards our work, and thank you for your detailed review. We are glad you found our work to be exceptionally presented, and our work to be high quality. To answer your question, the inspiration for this work came from one of the authors’ habit of watching live podcasts on YouTube. These are often hours long but have a still camera and a static background. Videos with large amounts of stationary content are prime candidates for methods like RLT - their visual information can be drastically compressed with minimal loss.

---

> > ### Comment · Reviewer_B8qa · 2024-08-12
> >
> > thank you for your response.

---

### Author Rebuttal · Authors · 2024-08-07

We thank all the reviewers for their insightful comments and are grateful for their feedback. We are glad they found our work to be __“commendably original, very well-motivated and intuitive”__ (RB8qa
, RN711, RtWGP), __“state-of-the-art, useful, significantly faster and better“__ (RB8qa, RN711,RtWGP) and __“exceptionally presented and well-written”__ (RB8qa, RtWGP). In this global rebuttal, we review the main contribution, address a common concern and outline the additional results provided in the PDF. We address each reviewer's individual concerns in the Author Rebuttals.

To reiterate, RLT provides a way to dramatically speed up video transformers by identifying redundant spatiotemporal patches before running the model. By removing those patches, we find that we can achieve significant speedups without loss of performance on inference for pre-trained models and during training. Our formulation of RLT avoids the common pitfall of having to use padding during training while devoting more computation to busier videos and more static ones.

__Concern (RN711,RtWGP): the results in Table 4 are confusing - why does the learned encoding hurt on ViT-L?__

We understand your concerns about the results on the learned length-encoding. The result in Table 4 of the main manuscript is a typo. RLT in fact achieves a top-1 accuracy of 84.0 when trained with the run-length encoding, an extremely small drop-off in performance. As we state in the text of the paper, and demonstrate in Table 2 of the results PDF, the results on ViT-L match the pattern of those of ViT-B, and show the learned encoding is especially helpful when combined with masking. We deeply apologize for this oversight on our part, and will be sure to update the manuscript accordingly.

__New Results__:

We include the results from several experiments requested by Reviewer N711 and Reviewer tWGP. Table 2 contains the updated results for the run-length encoding ablation with the correct numbers. Table 1 and 3 compare RLT to STA, another pruning baseline that can also apply to pre-trained models. Table 4 compares training RLT for the same number of samples vs the same number of tokens, and Table 5 shows the average sequence length produced by RLT across multiple datasets.

Given our responses to each reviewers’ concerns, we humbly request that you consider adjusting your scores, and look forward to a fruitful discussion period.

---

### Public Comment · ~Wenxuan_Huang2 · 2024-12-11
**Similar related work**

Good work! I would like to mention that this work is related to TokenExpansion [1], which uses token similarity to select and merge tokens to achieve training acceleration for transformers. Maybe it should be discussed.

[1] Huang, Wenxuan, et al. "A General and Efficient Training for Transformer via Token Expansion." Proceedings of the IEEE/CVF Conference on Computer Vision and Pattern Recognition. 2024.

---

### Public Comment · ~Zuowen_Wang2 · 2024-12-22

Congrats on the nice work!
It's nice to see work exploiting the input changes to reduce computations with ViT structures. We had a work [1] on this line in 2022, but we were targeting a different sensor named Event Camera. This sensor could be (overly simplified) considered as an input differencing sensor thus the goal was also to exploit the sparsity (or redundancy if it's frames) to accelerate inference.

[1] Exploiting Spatial Sparsity for Event Cameras with Visual Transformers, Zuowen Wang et al.

---

### Decision · Program_Chairs · 2024-09-25

**Decision:**

Accept (spotlight)

**Comment:**

This well-written paper introduces a novel method for reducing the computational burden of video analysis models.

The proposed method omits patches/tubelets that do not change significantly between frames, and uses a run-length embedding to capture the information about how much data is omitted. The result is a model that only needs to process the tokens that are changing in a video, which can lead to significant speedups depending on the video content, with little or no loss in performance on human action recognition tasks. The computational savings is greatest when the input video is static, but significant savings can be had in typical videos where the important activity take up only a small part of the total time, or where the motion activity is localized to a small area of the frame.

The method reduces static tokens in training time dynamically. RLT takes into account spatial changes with respect to the temporal aspect. The RLT method inspired by popular video compression techniques such as HEVC and AVC which are content aware, making RLT also content aware as part of the input embedding of the video transformer.

The rebuttal, and subsequent discussion, resolved most of the concerns of the initial reviews. One reviewer expressed concern that the method was "too simple". But, as the authors responded, this is not a weakness. Many important advances in ML have been at heart apparently simple ideas (e.g. "dropout"). In this case the contribution may perhaps be simple, but it is not trivial, and is worth being made known to the ML community.

The other main critique made by the reviewers had to do with the apparent loss in performance when RLT is used. However, the authors point out that there was a typo and the actual performance drop is minimal. It should be kept in mind that the main benefit of the approach is the significant speedup attained when running video transformer models.